

# Simulations of spaceborne multiwavelength lidar measurements and retrievals of aerosol microphysics

David N. Whiteman[1], Daniel Perez-Ramirez[2], Igor Veselovskii[3], Peter Colarco[1], Virginie Buchard[4],

[1]NASA/Goddard Space Flight Center, Greenbelt, MD, 20770, USA
[2]University of Granada, Spain
[3]University of Maryland, Baltimore County, Baltimore, MD, 21550 USA
[4]University of Maryland, Baltimore County, Baltimore, MD, 21550 USA

*Correspondence to*: David N. Whiteman (david.n.whiteman@nasa.gov)

**Abstract.** In support of the Aerosol, Clouds, Ecosystems missions, simulations of a spaceborne multiwavelength lidar are
performed based on global model simulations of the atmosphere along a satellite orbit track. The yield for aerosol
microphysical inversions is quantified and comparisons are made between the aerosol microphysics inherent in the global
model and those inverted from both the model's optical data and the simulated lidar measurements, which are based on the
model's optical data. We find that yield can be significantly increased if inversions based on reduced optical data are
acceptable. In general, retrieval performance is better for cases where the aerosol fine mode dominates. Lack of sensitivity to
coarse mode cases is found, in agreement with earlier studies. Surface area is generally the most robustly retrieved quantity.
The work here points toward the need for ancillary data to aid in the constraints of the lidar inversions and also for the need
for joint inversions involving lidar and polarimeter measurements.

## 1 Introduction

NASA's involvement in space lidar systems dates from the Apollo era when a laser altimeter was flown on Apollo 15, 16 and
17 in 1971-72 (http://www.lpi.usra.edu/lunar/missions/apollo/apollo_17/experiments/la/). Since that time, several NASA
missions have used laser altimeters for topographic measurements of both the Earth (Garvin et al., 1998) and other planets
(Smith et al., 2001). The first NASA lidar mission that measured the Earth's atmospheric profiles was the Lidar In space
Technology Experiment (LITE) that flew on board the space shuttle Discovery in 1994 for a 10-day technology
demonstration mission (McCormick et al, 1993). In 2006, the Cloud-Aerosol Lidar with Orthogonal Polarization (CALIOP)
lidar was launched as part of the Cloud-Aerosol Lidar and Infrared Pathfinder Satellite Observation (CALIPSO) mission and
continues to make profile measurements of aerosols and clouds into 2016.

A future NASA atmospheric profiling mission, the Aerosol, Cloud, Ecosystems (ACE) Mission, was identified as a priority
in the 2007 National Research Council Decadal Survey and is anticipated to include NASA's most advanced spaceborne
lidar system. The goals of the ACE mission include decreasing the uncertainties associated with the roles of aerosols, clouds
and precipitation in the hydrological cycle and climate change. The candidate suite of instruments proposed for the mission
includes polarimeter, radar, multiwavelength (MW) High Spectral Resolution Lidar (HSRL) and ocean color spectrometer.
NASA is conducting a pre-formulation study to consider science measurement requirements and measurement capabilities of



the candidate instrumentation. As a part of that pre-formulation study, we consider here the measurement capability of, and aerosol micro-physical retrievals from, a spaceborne, multiwavelength lidar within the context of the ACE mission. More information on the ACE mission along with the results of community workshops led by the ACE Science Working group can be found at http://dsm.gsfc.nasa.gov/ace/.

## 2 Aerosol microphysical retrievals using the multiwavelength lidar

The multiwavelength lidar technique has proven useful for estimating the vertical profile of aerosol microphysical parameters such as effective radius, volume and refractive index under widely varying conditions (Veselovskii et al., 2009, Ansmann et al., 2011, Müller et al., 2013, Veselovskii et al., 2015). The most common configuration of multiwavelength lidar for such studies is one based on a tripled Nd:YAG laser providing three backscattering and two extinction coefficients (the so called 3ß+2α dataset). From these data aerosol microphysics can be inverted (Müller et al., 1999 a, b, Veselovskii et al., 2002, Böckmann et al., 2005) using different inversion approaches (Donovan and Carswell, 1997, Veselovskii et al., 2012). It is most common for ground-based systems that Raman scattering is used for the measurements from which aerosol extinction is calculated. However, for spaceborne configurations the HSRL technique (Shipley et al., 1983, Rogers et al., 2009, Burton et al, 2014) will be considered in the simulations done here due to its much higher signal-to-noise for the pure molecular measurements. An important consideration in these lidar inversions is that there are only 5 input optical data and, thus, the inverse problem is under-determined (Veselovskii et al., 2005). The inverse equations are multi-valued implying that other information is needed to constrain the region over which a solution is sought. Such constraints increase the stability and accuracy of the inversion and need to be considered an important part of the solution technique. In the context of the ACE mission, additional information from a polarimeter or from lidar depolarization measurements (Burton et al., 2012) can also aid the lidar inversion by helping to constrain the search space for a solution although those details were not considered here. Additional assumptions about size- and spectrally-independent refractive index are also made in these retrievals (Müller et al., 1999 a,b; Veselovskii et al., 2002).

## 3 Hardware configuration

The main purpose of this effort was to perform a relatively simple study to assess the measurements and aerosol microphysical inversions from a spaceborne multiwavelength HSRL lidar. As such, we did not wish to consider the specific detailed optical design of any particular system that might be under development to meet the needs of the ACE mission. We also did not consider clouds. We focused only on the lidar system parameters that determine optical throughput and used them to simulate performance of the spaceborne lidar in a cloud-free atmosphere. To determine the system parameters, we consulted with various manufacturers and lidar specialists to generate the set of parameters shown in Table 1. These parameters are considered to be technically feasible and a reasonable representation of a currently feasible spaceborne HSRL lidar.

## 4 Simulation Approach

The model used in this simulation study is one that was first developed in 1999 (Whiteman et al., 2001) in support of the Raman Airborne Spectroscopic Lidar (Whiteman et al, 2007) development effort that was funded by NASA's Instrument Incubator Program. That model has been used to simulate both airborne and ground-based lidar systems under a wide range



of conditions (Whiteman et al., 2010). Modifications were made to the molecular cross sections used in the model to permit the simulation of the molecular channels of an HSRL instrument. For all simulated lidar signals, the model generates a profile of photon count rate based on the lidar instrument parameters and the atmospheric profile of aerosol and molecular density provided to the model. For validation of the model, consideration was given to using data from the CALIPSO (Winker et al., 2010) instrument. The CALIPSO lidar uses a 22-bit analog data acquisition system with gain switching capability so that the raw output is in digital count values that correlate to a voltage scale. To avoid the complications of converting this voltage value to photon count rate, we chose instead to use Geosciences Laser Altimetry System (GLAS) (Spinhirne et al., 2005) data from October 9, 2003 since the GLAS system used photon counting electronics. The measured photon count rate from GLAS could, therefore, be directly compared with the lidar simulator's output. Using GLAS instrument parameters, the comparisons of GLAS data with the model output were in excellent agreement including during the passage of the instrument into and out of the terminator region where skylight background is considerable. Beyond a photon count rate of 10-20 Mhz, however, the comparisons degraded due to the non-linear nature of the photon counting process of the GLAS electronics. This limitation did not influence the comparisons done here since the candidate ACE lidar is considered to use an analog detection system which does not possess the non-linear behavior shown in the GLAS photon counting data.

As the basis for the simulations, the Goddard Earth Observing System Model, Version 5 (GEOS-5, Rienecker et al. 2008) was used to simulate the atmospheric conditions along a 24-hr track of the CALIPSO lidar instrument from July 24, 2009. GEOS-5 was run using assimilated meteorology from its own Modern-Era Retrospective Analysis for Research and Applications (MERRA, Rienecker et al. 2011). Aerosols were run inline in GEOS-5 (Colarco et al. 2010) and included assimilation of MODIS-derived aerosol optical thickness (the so-called MERRAero aerosol reanalysis, Kessner et al. 2013; Buchard et al. 2014, 2015, 2016; Colarco et al. 2014a). The inline aerosol module is primarily a bulk scheme, with optical properties assigned as in Colarco et al. (2010), except for dust, where the treatment is for non-spherical particles as in Colarco et al. (2014b). We have found that the GEOS-5 based MERRAero fields used here are generally realistic in their representation of aerosol profiles with respect to CALIPSO observations (Buchard et al. 2015; Nowottnick et al. 2015).

Density and aerosol optical profiles from GEOS-5 were extracted along the CALIPSO track at a 70 km horizontal resolution and extending up to an altitude of approximately 80 km. These profiles were then used as input to the lidar simulator. The vertical resolution of the GEOS-5 profiles ranged from 0.12 km at the lowest levels of the atmosphere to approximately 4 km at the top of the profile. Scene brightness information was provided by the VLIDORT (Vector LInearized Discrete Ordinate Radiative Transfer) radiative transfer model (Spurr et al., 2006, Buchard et al., 2015). Using simulated atmospheric profiles along the CALIPSO track permits the optical measurements of the ACE candidate lidar to be compared with those of the CALIPSO lidar. Also, we were able to perform microphysical inversions using both the GEOS-5 optical data as well as the lidar simulations of the GEOS-5 optical data and compare with the aerosol microphysics inherent in the GEOS-5 model. GEOS-5 generates cloud fields but, as mentioned, they were ignored for the studies done here to simplify the assessment of basic lidar measurement capability.

A comparison of simulated ACE lidar backscatter and extinction measurements at 532 nm using the parameters listed in Table 1 (with orbital altitude of 450 km) and the GEOS-5 optical inputs is shown in Fig. 1. We chose to display the mid-visible wavelength results only to simplify the graph. The atmospheric conditions simulated by GEOS-5, and used as input to the numerical model, are those from 0102 UT on July 24, 2009 at a point over central Africa. This constitutes a nighttime simulation with the backscatter comparisons on the left and the extinction comparisons on the right. For both backscatter and extinction, the optical input data from GEOS-5 are shown ("GEOS532") along with two simulations of the lidar



measurements; the first without random uncertainty ("NoNoise") and the second with random uncertainties consistent with the lidar system hardware and the atmospheric conditions ("Sim532"). For both backscatter and extinction, the lidar

simulations without random uncertainty generally agree within 1-3% of the GEOS-5 inputs except for portions of the profiles showing significant gradients where the 150 m and 450 m vertical resolution of the backscatter and extinction profiles, respectively, can lead to larger differences. This indicates that the lidar numerical simulations can faithfully reproduce the input optical data. The lidar simulations that include random uncertainty indicate that for this atmospheric profile, the simulated lidar system would measure aerosol backscatter and extinction in the boundary layer with generally

less than 3% and 20% random uncertainty, respectively. Such results were found to be typical for a nighttime simulation.

With the lidar simulation model validated by comparison with GLAS data and shown to reliably reproduce the GEOS-5 optical inputs as shown in Fig. 1, a comparison was then made of the lidar-simulated measurements at 532 nm and the actual 532 nm CALIPSO lidar data from July 24, 2009 under both day and night conditions. Examples of such comparisons are given in Fig. 2, where daytime profiles are shown on the left and nighttime on the right. Because of the difficulty to

determine photon flux from the CALIPSO data, the simulated lidar profiles were normalized in the region above the boundary layer. Considering the portions of the profiles above the boundary layer, we estimate that a spaceborne lidar system with the specifications shown in Table 1 would possess a factor of approximately 3-5 higher signal to noise backscatter measurements under nighttime conditions than the CALIPSO lidar and more than an order of magnitude higher signal to noise in the daytime. The mismatch in the height of the planetary boundary layer between the lidar simulations and the actual

CALIPSO lidar measurements shown is not surprising given that boundary layer height is a difficult quantity to forecast in atmospheric models in general. It is particularly true for a global simulation as was used for these studies.

## 5 Inversion Schemes and Input Data Used

One of our goals was to study the inversion of simulated spaceborne multiwavelength lidar data to microphysical quantities such as effective radius, concentration (volume, surface) and index of refraction. We considered two different inversion

techniques, that of regularization (Twomey, 1977, Müller et al., 1999 a,b, Veselovskii et al., 2002) and linear estimation (Twomey, 1977, Donovan and Carswell, 1997, Veselovski et al., 2012, De Graaf et al., 2013). The regularization technique uses a linear combination of triangular basis functions in the inversion and, when used with 3 ß+2α optical input data, results in estimates of both the volumetric quantities, such as effective radius and concentration, and the particle size distribution (PSD). The inversion based on the linear estimation (LE) approach expands the solution directly in terms of the Mie kernels

in order to retrieve volumetric quantities but does not provide a reasonable estimate of PSD. Previous research (Veselovskii et al, 2012) indicates that reasonable estimates of volumetric quantities are achievable using the LE technique and 3 ß+1α optical input data. Also, the LE technique is computationally much more efficient than the regularization technique, although computations using the regularization technique can be made more efficient through the use of pre-computed tables of Mie scattering values.

In the study here, we performed inversions based both on 3ß+2α input optical data as well as reduced, 3ß+1α optical data using both the regularization and LE techniques. For the 3ß+1α studies, we removed the 355 nm extinction measurement leaving just the 532 nm extinction measurement. Whether using regularization or LE with either 5 or 4 optical data as input, it must be remembered that the inversion problem is significantly under-determined implying that both the method of solution and the constraints used in the inversion influence the results obtained. Ancillary data that can help to decrease the

ranges of allowed radii and refractive index in the search for a solution can improve the solution obtained. Because of the





multi-valued solution space, in either the regularization or LE approaches, a family of solutions is obtained. The final solution is determined from this family by averaging ~1% of the solutions nearest to the minimum of discrepancy, where discrepancy refers to the difference between the input optical data and the optical data consistent with the inverted microphysical parameters (Veselovskii et al., 2002).

## 6 Results

Here we consider what fraction of the simulated lidar measurements would be of sufficient quality to support an inversion to aerosol microphysics. We consider how the inversions of aerosol microphysics, both for the original GEOS-5 optical data and for the simulated lidar measurements, compare with the aerosol microphysics inherent in the GEOS-5 model.

### 6.1 Yield

According to the draft ACE report from 2010 (http://acemission.gsfc.nasa.gov/documents/Draft_ACE_Report2010%20.pdf), the ACE lidar data, in some cases aided by or in combination with daytime polarimeter data, will be used to retrieve vertical profile of aerosol extinction and backscatter, vertical profile of effective radius and width of size distribution, vertical profile of refractive index, vertical profile of single scatter albedo and vertical profile of number concentration. Under nighttime conditions, only the lidar will be available to retrieve these parameters so it is important to study the ability of the lidar to perform microphysical retrievals without ancillary measurements.

The draft ACE report states that 3 backscatter and 2 extinction coefficients are needed with 15% accuracy from the multiwavelength lidar to support inversions of the lidar data to aerosol microphysical parameters. A minimum extinction threshold of 0.02 km$^{-1}$ at 532 nm was also established to support inversions. The 15% threshold is consistent with publications that have studied the influence of various amounts of random uncertainty on the inversions (Veselovskii et al., 2012, Perez-Ramirez et al, 2013).

We would now like to calculate the "yield" of a multiwavelength lidar with the specifications shown in Table 1 and considering the minimum extinction threshold of 0.02 km$^{-1}$ and maximum of 15% random uncertainty. To do so, we consider the 24-hr dataset from GEOS-5 and "bins" of the atmosphere measuring 450 m in the vertical dimension and 80 km horizontally. If a bin has average extinction at 532 nm greater than 0.02 km$^{-1}$ and the lidar simulations indicate that all 5 optical data (3ß+2α) possess less than 15% random uncertainty for the bin, then measurements of that bin would support a 3ß+2α-based microphysical inversion. We also consider the same question but for the 3ß+1α dataset, where we exclude the 355 extinction measurement. We perform this yield analysis for the two orbital altitudes of 450 km and 820 km. The lower altitude is preferred from the standpoint of maximizing the signal-to-noise of a spaceborne lidar system, while the higher altitude would permit the spaceborne lidar to fly in formation with the A-train constellation, which includes Terra, Aqua and other satellites, thus increasing possibilities of data fusion from different spaceborne sensors. In these yield studies, it must be remembered that cloud-free simulations from GEOS-5 have been considered. We estimate later the influence of a cloudy atmosphere on these results.

Tables 2 and 3 provide the results of the yield studies for 3ß+2α and 3ß+1α retrievals where we have considered only retrievals over land and where various thicknesses of the lower atmosphere are studied. Land-based retrievals were used here to focus on the larger aerosol events that occur over land and are most likely to be the ones of interest in spaceborne lidar



studies. Considering the lowest 3, 4, 5 km of the 8640 profiles in the GEOS-5 simulation indicates that 36%, 31% and 27%, respectively, of the bins over land have extinction at 532 nm of 0.02 km$^{-1}$ or greater. These are the bins considered to "qualify" for a microphysical inversion if the suite of multiwavelength lidar measurements of that bin are of sufficient quality. From an orbital altitude of 450 km, and considering the 3ß+2α (3ß+1α) measurement suite, approximately 15% (35%) of those qualifying bins would be measured with uncertainties of 15% or better. The large increase in the number of bins measured with 15% uncertainty when considering only 3ß+1α is due to the fact that the measurement deleted is the 355 nm extinction, which is the one that possesses the highest noise in the simulations. From an orbital altitude of 820 km, and considering 3ß+2α (3ß+1α) measurements, approximately 3% (15%) of the qualifying bins are measured with 15% uncertainty or better. One should realize that while the 3ß+1α based inversions provide a higher yield, the information content of those inversions will be less than those based on 3ß+2α.

**6.2 Inversion Comparison**

GEOS-5 contains information that either directly or indirectly can be converted to aerosol microphysical parameters. The aerosol module is based on the Goddard Chemistry, Aerosol, Radiation, and Transport (GOCART) model (Chin et al. 2002; Colarco et al. 2010). The GOCART module carries aerosol mass in a series of tracers: 5 tracers representing size of dust, 5 tracers representing size of sea salt, 2 tracers each of black and organic carbon (one for hydrophobic and one for hydrophilic modes) and 1 tracer for sulfate. The conversion of the resolved species masses to optical quantities is done through pre-computed lookup tables, where for each tracer a dry particle size distribution is assumed, along with a hygroscopic growth factor, which is a function of relative humidity (hygroscopic growth is relevant for sulfate, sea salt, and hydrophilic carbon, and is not considered for dust and hydrophobic carbon). For dust and sea salt, the particle size distribution is explicitly resolved. Spectral refractive indices are input for generating the lookup tables, primarily from the OPAC (Optical Properties of Aerosols and Clouds) database (Hess et al. 1998), and the lookup tables are calculated assuming Mie theory. The exception is for dust, where a spectrally varying but weakly absorbing dust refractive index is assumed and particle non-sphericity is considered for determining optical properties using the database from Meng et al. (2010) (see also Colarco et al. 2014b; Buchard et al. 2015). Further details are provided in Colarco et al. (2010). The lookup tables therefore establish the linkage between the bulk mass simulated, the assumed particle microphysical properties, and the optical quantities input to the lidar retrievals.

In the studies done here, we consider the GEOS-5 aerosol microphysics as the baseline for comparing the different inversions of optical data. For these inversions, we consider both regularization and linear estimation techniques and different sets of input optical data: 1) the optical data contained in GEOS-5 and 2) the simulated lidar optical data. For both of these optical data sources we consider 3ß+2α and 3ß+1α. We perform the inversions and then make comparisons of effective radius, volume, surface area and index of refraction. For the inversions done here, we limited the range of imaginary refractive index to $m_I <= 0.01$, or low absorbing particles, due to the increase in uncertainty for retrievals of more highly absorbing particles shown in earlier studies (Veselovskii et al., 2005). All qualifying cases were separated into fine, mixed and coarse aerosol modes using the fine mode fraction as determined using the spectral deconvolution technique (O'Neil et al., 2003) applied to the GEOS-5 optical data. Based on this size classification, the radius constraints were determined as shown below in a similar manner as done in Pérez-Ramírez et al., (2015). For all cases, the real part of the index of refraction was allowed to vary between 1.35 – 1.65 in steps of 0.0025 and the imaginary index varied between 0.001 and 0.01 with steps of 0.001.





Fine Mode Dominant: $\eta > 0.75$, $R_{min} = 0.075$ and $R_{max} = 2$ um

Mixture of Modes: $0.25 < \eta < 0.75$, $R_{min} = 0.075$ and $R_{max} = 10$ um

Coarse Mode Dominant: $\eta < 0.25$, $R_{min} = 0.3$ and $R_{max} = 10$ um

Using the above inversion schemes and input datasets, examples of aerosol volume comparisons are given in Fig. 3, where both fine mode cases (2098 qualifying cases with random uncertainty less than 15% in all optical channels as done in Table 225 2) and coarse mode cases (#=1439) are shown. In this figure, the volume directly from GEOS-5 is shown along with the inverted volume using various combinations of data and inversion scheme as follows:

1. Using simulated Lidar optical data with GEOS-5 optical data as input to the simulator
    1. Regularization with 3ß+2α
    2. Regularization with 3ß+1α
3. Linear Estimation with 3ß+2α
    4. Linear Estimation with 3ß+1α
2. Using GEOS-5 optical data directly
    1. Regularization with 3ß+2α
    2. Regularization with 3ß+1α
3. Linear Estimation with 3ß+2α
    4. Linear Estimation with 3ß+1α

To be clear, the GEOS-5 optical data are the ones that are used as inputs to the lidar simulator model and are therefore inherently noise-free. An example of the GEOS-5 profile data used in these simulations and inversions is given as the 240 "GEOS532" profile in Fig. 1. In Figs. 3 and 4, only qualifying cases have been included.

Figure 3 shows the comparisons of histograms of aerosol volume retrievals using all the combinations of data and inversion techniques shown above for the cases (left) where the fine mode fraction is greater than 0.75; indicative of fine mode cases and (right) where the fine mode fraction is > 0.25 indicative of coarse mode cases. Aerosol volume has proven to be one of the more robustly retrievable microphysical quantities using the multiwavelength lidar technique due to a generally good 245 correlation between the measured extinction and particle volume. In these comparisons, we take the GEOS-5 microphysical data as the reference and compare the histograms of the inverted microphysical data to that of GEOS-5. In general, for the fine mode cases shown in Fig. 3, the different volume retrievals agree well with each other and with GEOS-5 although there is a tendency for the inversions to show a larger number of small volume cases (c.f. the peak around 15-20 $\mu m^3$ $cm^{-3}$) and a smaller number of larger volume cases (c.f in the vicinity of 100 $\mu m^3$ $cm^{-3}$) than the GEOS-5 reference. On the right of Fig. 3 250 is shown the comparison of coarse mode cases where it is apparent that the histograms of all inversion schemes show considerable disagreement with the GEOS-5 reference.

We investigate these disagreements further by considering the histograms of effective radius shown in Fig. 4. The fine mode cases are shown on the left and the coarse mode cases on the right. We consider first the fine mode cases shown on the left. Close inspection of these cases indicates that the 3ß+2α based retrievals are better able to retrieve particle sizes down to 0.1 255 μm but that all retrieval schemes under-represent these small particles, consistent with the small particle insensitivity revealed in earlier information content studies (Veselovskii et al., 2005). The situation is considerably worse when



considering 3ß+1α based retrievals with all fine mode case retrievals performing poorly in representing small particles although the regularization technique does show more skill in this respect. The loss of the 355 nm extinction measurement clearly compromises the ability to retrieve the smallest particles sizes. On the right of Fig. 4 is shown the comparison of

coarse mode cases where it is clear that no retrieval scheme is able to capture the sizes associated with the sea salt aerosols that are present in GEOS-5. Figure 4 can be used to better understand the degradation of agreement between the retrieved volumes and the GEOS-5 histograms shown in Fig. 3 as particle size increases.

The preceding discussion concerning volume and effective radius retrievals has been qualitative in nature. To better quantify the performance of the various retrieval schemes with respect to the GEOS-5 reference we will quantify the comparison of

the different retrieval schemes in the following ways. For all parameters except refractive index, a root-mean-square deviation metric is calculated as a percentage as

$$100 \frac{1}{N} \frac{\sum \sqrt{\left( X_{GEOS} - X_{Ret} \right)^2}}{X_{GEOS}} \tag{1}$$

where $X_{GEOS}$ is the reference microphysical quantity from GEOS-5 and $X_{Ret}$ is the inverted microphysical value using one of the inversion schemes under consideration. For the refractive index comparisons, we calculate the fractional deviation

metric as

$$\frac{1}{N} \sum \sqrt{\left( X_{GEOS} - X_{Ret} \right)^2} \tag{2}$$

Equations (1) and (2) were evaluated for each individual retrieval and the composite values presented in Tables 4 and 5. Deviations are color-coded in green, yellow and red based on the magnitude of the deviation metric and consideration of the desired uncertainty in the retrieved quantities discussed in the ACE draft report of 2010. The color coding scheme is given in

Table 6, where green indicates values fully consistent with the desired uncertainties expressed in the draft report, yellow indicates marginal consistency, while red indicates uncertainties inconsistent with desired uncertainties.

Table 4 considers the deviation between inversions based on GEOS-5 optical data and the microphysics inherent in GEOS-5. To be clear, the profiles used in these inversions are not generated by the lidar simulator but rather are the ones that come from GEOS-5 and are used as input to the lidar simulator. The results shown in Table 5, by contrast, are based on inversions

of simulated lidar data with varying amounts of random uncertainty. The results shown in Table 4 can be considered as a type of information content study since there are no random errors in the input optical data; the data are derived from GEOS-5 microphysics using Mie scattering codes. Table 5 considers inversion results based on the simulated lidar profiles that use as their input the GEOS-5 optical profiles that were used to generate Table 4. Table 5 therefore quantifies the retrieval capability of the candidate spaceborne lidar using the different inversion approaches considered while the comparison of

Tables 4 and 5 can be used to study the influence of random errors on the various retrieval techniques.

For the fine mode cases shown in Table 4, the regularization retrieval using 3ß+2α input optical data agrees well with the GEOS-5 microphysics for all quantities, except for the retrieved imaginary index which is slightly outside of the desired bounds. The other retrieval schemes show progressively poorer agreement with the GEOS-5 microphysics as you consider regularization (3ß+1α), LE (3ß+2α), LE (3ß+1α). For a mixture of particles, regularization with either 3ß+2α or 3ß+1α input

data perform similarly and generally as well as each other except that the real part of the refractive index shows considerable deviation from the reference. The imaginary index is closer to the desired values, though, than for the fine mode cases.



Again, as in the fine mode comparisons, the linear estimation retrievals do not perform as well for these mixed mode cases as those based on regularization. Considering the coarse mode cases, only surface area, which is highly correlated with the optical extinction values, can be retrieved by any of the retrieval schemes with good accuracy. In general, Table 4 indicates that the regularization technique shows more skill in inverting these noise-free data into the desired microphysical information and performs well for fine and mixed mode cases.

Table 5 provides the comparison of the reference GEOS-5 microphysics with the inversions based on simulated lidar data for the system specifications shown in Table 1 and a 450 km orbital altitude. Figure 1 contains examples of the input data used for these inversion studies. A quick look at Table 5 indicates that these inversions based on simulated lidar data show larger deviations from the GEOS-5 microphysics than the results shown in Table 4 based on noise-free data. For fine mode cases, effective radius retrievals show considerable deviation from the GEOS-5 reference although volume is retrieved well when using the regularization technique. It should be noted that these retrievals of volume are essentially insensitive to random uncertainties of up to 50%. For the case of mixed mode retrievals using simulated lidar data, the most conspicuous result is that the regularization technique coupled with 3ß+1α optical data shows considerable skill at retrieving all quantities except the real part of the refractive index. These comparisons degrade only slightly for increasing random uncertainty up to 50%. For coarse mode cases, retrievals of surface area are very robust for all inversion techniques and for random uncertainties up to 50% although the LE based retrievals consistently show slightly smaller deviations than those of regularization.

## 7 Discussion and Conclusions

Our work indicates that the spaceborne lidar simulated here would possess substantially improved backscatter measurements over previous spaceborne lidars such as GLAS and CALIPSO. Although not tested here, these higher signal-to-noise measurements should improve the ability to retrieve boundary layer heights and to detect thin cirrus. The addition of pure molecular measurements through the HSRL technique will permit direct measurements of aerosol optical thickness and extinction. The combination of backscatter and extinction measurements permits retrievals of aerosol microphysical quantities as shown here.

The availability of atmospheric profiles along a 24-hr satellite track from GEOS-5 has permitted us to assess the yield of a spaceborne lidar under realistic conditions globally and with large statistics. In this effort, we defined yield based on a maximum uncertainty of 15% for aerosol extinction values in excess of 0.02 km$^{-1}$, although some results in this paper indicate that inversions can tolerate considerably higher amounts of random uncertainty for certain quantities such as 1) surface area for mixed and coarse mode aerosols and 2) volume for fine mode aerosols. The yield from a 450 km orbit, not accounting for clouds, is approximately 4-5% when considering the use of 3ß+2α input data. Yields increase by approximately a factor of 2.5 when considering inversions based on 3ß+1α input data. Inversions based on 3ß+1α input data support only estimates of volumetric quantities, however. There are instances, such as for mixed mode aerosols, where retrievals based on 3ß+1α show higher skill than those based on 3ß+2α and the retrievals are highly resistant to input uncertainty. Yields in such cases will be considerably higher than shown in Tables 2 and 3. For reasonable estimates of the particle size distribution from lidar, however, the full 3ß+2α input dataset is needed along with the regularization technique. The increased resistance to uncertainty when inverting the 3ß+1α dataset can be understood from Perez-Ramirez et al., (2013) where it was shown that biases in the input extinction data cause larger deviations in the retrieved parameters than the same uncertainty in input backscatter data would produce. Therefore, elimination of the 355 nm from the input dataset results in a more stable inversion that can tolerate higher amounts of input noise.





It is no surprise that the yields from an orbital altitude of 820 km are lower than from 450 km. From 820 km, using the full 3ß+2α dataset, yields are approximately 1%. The use of the 3ß+1α dataset results in a yield increase of a factor of 5 from 820 km. Again, the high resistance of certain retrievals to uncertainty in the input data imply that real yields for certain quantities would be higher than those shown in Tables 2 and 3. To simplify data analysis, this study ignored the cloud fields present in GEOS-5. Estimates of cloud abundance based on the CALIPSO data from July 24, 2009 indicate that yields for aerosol
inversions below 5 km, as provided in Tables 2 and 3, would be reduced by a factor of approximately 5-10 if clouds had been included in the analysis.

Here we have constrained the retrievals based on the fine mode fraction as determined using the spectral deconvolution technique. In under-determined problems such as this, the selection of appropriate constraints is an important part of the solution technique and we did not make use of the depolarization measurements, which are expected to be available on an
ACE lidar system, in determining the constraints. Use of such depolarization data to aid aerosol typing (Burton et al., 2014) could further narrow the constraints used and potentially improve the inversion results. Consideration of the backscattering Ångström exponent can also add information that can be used for constraint (Veselovskii et al., 2015). It is expected, however, that with 3ß+2α optical data there will be some lack of sensitivity to both the smallest and largest particles in naturally occurring aerosol size distributions (Veselovskii et al., 2005) and the results presented here reflect that lack of
sensitivity. We also have only considered low absorbing solutions ($m_I$ < 0.01) since previous work has also indicated increased inversion uncertainties for more highly absorbing particles (Veselovskii et al., 2005). We avoided those difficulties by not considering solutions with $m_I$ > 0.01.

The use of a 24-hour atmospheric simulation from GEOS-5 along with the reference aerosol microphysics values from GEOS-5 provided a large set of statistics for comparison of techniques. Those statistics are summarized in Tables 4 and 5.
The conclusions that can be drawn from the results shown in these tables are as follows:

1.  Effective radius was well retrieved for fine mode distributions if the optical data were noise-free but the addition of random uncertainty up to 15% to the input optical data degraded the retrieval of effective radius significantly. A lack of sensitivity to particles with sizes less than 0.1 μm was demonstrated.

2.  Overall, surface area was the most easily retrieved quantity with good retrievals occurring, depending on inversion
technique, for random uncertainties up to 50%.

3.  For coarse mode aerosols, only surface area retrievals were reliable. A lack of sensitivity to particle sizes of ~2.0 μm and greater was demonstrated.

4.  Retrievals showed resistance to random errors up to 50% for the following cases:

    1.  fine mode volume retrievals

2.  mixed mode V, S, $m_I$ retrievals

    3.  coarse mode surface concentration

5.  The regularization technique, in general, performed better than the linear estimation technique with LE performing slightly better than regularization only for the coarse mode surface area retrievals

The degraded performance of the lidar-based inversions for mixed mode and coarse mode cases is related to the decreased information content for large particles as previously stated. Also, these larger particles include cases of irregularly shaped dust which cannot be accurately described by the Mie kernels used in these inversions. Spheroidal models have been used for





inversion of lidar data in the case of dust aerosols (Veselovskii et al., 2010) although there remains uncertainty in how well the spheroidal model captures the scattering properties of various irregular aerosol types.

The work presented here has been done in the context of the ACE decadal survey mission. We have studied the aerosol microphysical retrieval capability of a candidate lidar without input from other data sources. Such information is useful to understand the inherent capabilities of a spaceborne lidar given that during the nighttime, lidar retrievals may be the only source of certain aerosol microphysical information. The work done here is hardly an exhaustive study of the ability to invert aerosol microphysics from spaceborne lidar data, however. Rather it should be considered an initial study that can help to

point toward needed refinements. For example, as mentioned above, the retrieval of effective radius for fine mode cases degraded significantly between the GEOS-5 based inversions and the lidar based ones that used data with 0-15% random uncertainty. By additional studies, we found that either constraining the maximum radius to no larger than 1 micron or further limiting the maximum random uncertainty brought the inversion results into much better agreement with the GEOS-5 reference. In this way, the tables point toward other inversion sensitivity studies that would be useful for determining

optimized constraints. In this process, the question arises for how to tighten the constraints used in the inversions in actual practice. In order to do so, other methods of constraint based on aerosol typing or the use of depolarization data and spheroidal kernels can be studied. Such studies will add useful information and help to determine an optimized approach for inverting multiwavelength lidar data in the absence of ancillary data as will be needed during the nighttime for a space mission. During the daytime, however, there should be other sources of data such as the anticipated polarimeter instrument,

from which column-averaged aerosol microphysical information can be determined (Mischenko et al., 1997, Dubovik et al., 2011). While additional work remains in refining the approach for optimal retrieval of aerosols from lidar alone as done here, initial promising steps (Liu et al., 2015) are being taken to develop joint inversions that couple lidar and polarimeter data. Much future work should be focused in this new area as well.

**Acknowledgement**

This work was supported by NASA Headquarters are part of the ACE pre-formulation study. It has also been supported by the Marie Skłodowska-Curie Individual Fellowships (IF) ACE_GFAT (grant agreement No 659398). The authors gratefully acknowledge several fruitful discussions and exchanges with Chris Hostetler, Rich Ferrare, Jon Hair, Kathy Powell, Detlef Müller and Sharon Burton of NASA/Langley Research Center.

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



Tables

| Simulated Lidar System Specifications | |
|---|---|
| Component/Parameter | Specification |
| Laser Power (1064, 532, 355 nm) | 10W, 10W, 5W |
| Laser Repetition Rate | 100 Hz |
| Telescope Diameter | 1.5m |
| Telescope Field of View | 130 micro radian |
| Channel Bandwidths (1064, 532, 355 nm) | 30 pm, 30 pm, 20 pm |
| Total optical efficiencies (molecular, particle) | 9.6%, 2.4% |
| Data Acquisition Technique | Analog |
| Orbital Altitudes Studied | 450 km, 820 km |

Table 1. Specifications of the spaceborne lidar system used in the simulation study.

| Yield from 450 km | | | |
|---|---|---|---|
| A bin is 450 m in the vertical and 80 km in the horizontal | Fraction of bins with extinction >= 0.02 km$^{-1}$ @ 532 nm (i.e. "qualifying" bin) | Fraction of "qualifying" bins measured with <= 15% uncertainty (3+2 inversions) | Fraction of "qualifying" bins measured with <= 15% uncertainty (3+1 inversions) |
| Below 3 km | 36% | 15% | 36% |
| Below 4 km | 31% | 14% | 35% |
| Below 5 km | 27% | 14% | 35% |

Table 2. Yield from an orbital altitude of 450 km. Information is given on what fraction of bins in the GEOS-5 simulations possess sufficient extinction for inversion and then what fraction of those bins are measured with the desired uncertainty.



| A bin is 450 m in the vertical and 80 km in the horizontal | Yield from 820 km | | |
|---|---|---|---|
| | Fraction of bins with extinction >= 0.02 km$^{-1}$ @ 532 nm (i.e. "qualifying" bin) | Fraction of "qualifying" bins measured with <= 15% uncertainty (3+2 inversions) | Fraction of "qualifying" bins measured with <= 15% uncertainty (3+1 inversions) |
| Below 3 km | 36% | 3.0% | 15% |
| Below 4 km | 31% | 3.1% | 15% |
| Below 5 km | 27% | 2.9% | 14% |

Table 3. Yield from an orbital altitude of 820 km. Information is given on what fraction of bins in the GEOS-5 simulations possess sufficient extinction for inversion and then what fraction of those bins are measured with the desired uncertainty.



| | Comparisons Using GEOS-5 Optical Data for Inversions | | | | | | | | | | | |
|---|---|---|---|---|---|---|---|---|---|---|---|---|
| | FINE MODE (η > 0.75) | | | | MIXTURE (0.25 < η < 0.75) | | | | COARSE MODE (η < 0.25) | | | |
| | Reg 3b2a | Reg 3b1a 532 | LE 3b2a | LE 3b1a 532 | Reg 3b2a 532 | Reg 3b1a 532 | LE 3b2a | LE 3b1a 532 | Reg 3b2a | Reg 3b1a 532 | LE 3b2a | LE 3b1a 532 |
| $R_{eff}$ | 23.2 | 51.1 | 26.7 | 66.1 | 13.0 | 22.0 | 41.4 | 22.9 | 54.0 | 65.1 | 69.2 | 68.5 |
| V | 12.4 | 16.7 | 21.5 | 31.9 | 14.5 | 14.1 | 41.6 | 36.1 | 54.5 | 67.8 | 71.4 | 72.0 |
| S | 21.4 | 39.8 | 30.1 | 53.5 | 11.5 | 14.4 | 12.4 | 19.2 | 5.7 | 16.2 | 9.9 | 12.7 |
| $m_r$ | 0.02 | 0.02 | 0.03 | 0.03 | 0.13 | 0.13 | 0.08 | 0.08 | 0.08 | 0.15 | 0.09 | 0.10 |
| $m_i$ | 9E-3 | 8E-3 | 9E-3 | 8E-3 | 2E-3 | 2E-3 | 5E-3 | 4E-3 | 3E-3 | 3E-3 | 4E-3 | 4E-3 |
| # of data | 3378 | | | | 1851 | | | | 2383 | | | |

Table 4: Comparison of GEOS-5 aerosol microphysical data and inversions of microphysical data using different sets of GEOS-5 optical data and both regularization and linear estimation inversion techniques. The values shown come from evaluation of the root-mean-square and fractional metrics that are defined in eqs. 1 and 2 in the text. In the column headings the type of retrieval scheme used is indicated (regularization or linear estimation) as well as the combination of optical input data. 3b2a means that the full 3ß+2α optical input data have been used, while 3b1a532 indicates that only 1 extinction value was used and it was at 532 nm.





| | Comparisons Using Simulated Lidar Data for Inversions | | | | | | | | | | | | | | | | | | | |
|---|---|---|---|---|---|---|---|---|---|---|---|---|---|---|---|---|---|---|---|---|
| | **Case A: η > 0.75 (Fine Mode Predominance)** | | | | | | | | | | | | | | | | | | | |
| | Uncertainties 0-15 % | | | | Uncertainties 15-20 % | | | | Uncertainties 20-30 % | | | | Uncertainties 30-40 % | | | | Uncertainties 40-50 % | | | |
| | Reg 3b2a | Reg. 3b1a 532 | LE 3b2a | LE 3b1 a532 | Reg 3b2a | Reg. 3b1a 532 | LE 3b2a | LE 3b1a 532 | Reg 3b2a | Reg. 3b1a 532 | LE 3b2a | LE 3b1a 532 | Reg 3b2a | Reg. 3b1a 532 | LE 3b2a | LE 3b1a 532 | Reg 3b2a | Reg. 3b1a 532 | LE 3b2a | LE 3b1a 532 |
| $R_{eff}$ | 48.6 | 59.3 | 44.9 | 73.3 | 51.2 | 53.2 | 45.6 | 64.9 | 54.4 | 55.2 | 49.7 | 67.2 | 53.2 | 56.4 | 47.3 | 68.3 | 48.9 | 59.8 | 43.1 | 61.2 |
| V | 16.2 | 18.4 | 22.1 | 30.5 | 16.7 | 17.7 | 21.5 | 28.4 | 19.6 | 19.4 | 23.5 | 28.4 | 17.7 | 17.6 | 21.2 | 26.2 | 16.5 | 17.3 | 20.1 | 25.3 |
| S | 34.3 | 39.3 | 35.3 | 52.3 | 33.7 | 37.3 | 35.0 | 49.3 | 37.0 | 38.7 | 34.5 | 48.9 | 35.5 | 37.0 | 32.0 | 47.2 | 36.0 | 35.9 | 28.2 | 43.4 |
| $m_r$ | 0.03 | 0.03 | 0.04 | 0.03 | 0.03 | 0.03 | 0.04 | 0.03 | 0.04 | 0.03 | 0.04 | 0.03 | 0.04 | 0.03 | 0.04 | 0.03 | 0.04 | 0.04 | 0.05 | 0.03 |
| $m_i$ | 9E-3 | 8E-3 | 8E-3 | 8E-3 | 8E-3 | 8E-3 | 8E-3 | 7E-3 | 8E-3 | 8E-3 | 7E-3 | 7E-3 | 8E-3 | 8E-3 | 8E-3 | 8E-3 | 7E-3 | 7E-3 | 7E-3 | 7E-3 |
| | **Case B: 0.25 < η < 0.75 (Mixture)** | | | | | | | | | | | | | | | | | | | |
| | Uncertainties 0-15 % | | | | Uncertainties 15-20 % | | | | Uncertainties 20-30 % | | | | Uncertainties 30-40 % | | | | Uncertainties 40-50 % | | | |
| | Reg. | Reg. 3b1a 532 | LE | LE 3b1 a532 | Reg. | Reg. 3b1a 532 | LE | LE 3b1a 532 | Reg. | Reg. 3b1a 532 | LE | LE 3b1a 532 | Reg. | Reg. 3b1a 532 | LE | LE 3b1a 532 | Reg. | Reg. 3b1a 532 | LE | LE 3b1a 532 |
| $R_{eff}$ | 40.5 | 22.6 | 52.1 | 23.0 | 42.4 | 24.4 | 52.3 | 24.8 | 44.4 | 24.2 | 54.2 | 25.6 | 52.0 | 28.2 | 58.2 | 25.4 | 52.1 | 33.1 | 57.8 | 25.8 |
| V | 28.1 | 19.9 | 52.5 | 40.1 | 30.7 | 22.0 | 52.0 | 39.7 | 32.1 | 22.6 | 53.7 | 40.7 | 37.0 | 23.5 | 57.2 | 44.0 | 36.6 | 28.8 | 56.5 | 39.8 |
| S | 43.6 | 22.1 | 16.0 | 26.2 | 46.7 | 23.4 | 17.0 | 27.3 | 49.7 | 23.8 | 17.4 | 27.6 | 60.0 | 24.9 | 19.1 | 29.5 | 67.8 | 25.6 | 21.9 | 28.8 |
| $m_r$ | 0.11 | 0.13 | 0.08 | 0.08 | 0.11 | 0.13 | 0.08 | 0.08 | 0.10 | 0.12 | 0.08 | 0.08 | 0.11 | 0.13 | 0.09 | 0.08 | 0.11 | 0.12 | 0.09 | 0.08 |
| $m_i$ | 2E-3 | 2E-3 | 5E-3 | 4E-3 | 2E-3 | 2E-3 | 5E-3 | 4E-3 | 3E-3 | 2E-3 | 5E-3 | 4E-3 | 2E-3 | 2E-3 | 5E-3 | 4E-3 | 2E-3 | 2E-3 | 5E-3 | 4E-3 |
| | **Case C: η < 0.25 (Coarse Mode Predominance)** | | | | | | | | | | | | | | | | | | | |
| | Uncertainties 0-15 % | | | | Uncertainties 15-20 % | | | | Uncertainties 20-30 % | | | | Uncertainties 30-40 % | | | | Uncertainties 40-50 % | | | |
| | Reg. | Reg. 3b1a 532 | LE | LE 3b1 a532 | Reg. | Reg. 3b1a 532 | LE | LE 3b1a 532 | Reg. | Reg. 3b1a 532 | LE | LE 3b1a 532 | Reg. | Reg. 3b1a 532 | LE | LE 3b1a 532 | Reg. | Reg. 3b1a 532 | LE | LE 3b1a 532 |
| $R_{eff}$ | 58.2 | 65.6 | 70.0 | 68.4 | 59.2 | 65.8 | 70.2 | 68.1 | 55.9 | 64.8 | 70.3 | 69.2 | 55.3 | 65.1 | 70.8 | 70.2 | 49.1 | 67.6 | 70.9 | 72.1 |
| V | 60.7 | 68.3 | 71.7 | 71.8 | 61.2 | 67.2 | 71.5 | 71.2 | 57.6 | 67.5 | 71.3 | 72.5 | 56.7 | 69.3 | 72.5 | 74.7 | 50.9 | 72.1 | 71.1 | 77.5 |
| S | 14.7 | 21.8 | 14.4 | 17.8 | 14.9 | 20.2 | 14.0 | 16.3 | 15.2 | 21.1 | 14.9 | 17.7 | 12.5 | 21.3 | 13.1 | 19.0 | 12.0 | 22.1 | 11.5 | 20.5 |
| $m_r$ | 0.14 | 0.15 | 0.11 | 0.11 | 0.14 | 0.15 | 0.11 | 0.11 | 0.15 | 0.15 | 0.11 | 0.11 | 0.15 | 0.15 | 0.11 | 0.12 | 0.12 | 0.23 | 0.12 | 0.22 |
| $m_i$ | 4E-3 | 3E-3 | 3E-3 | 3E-3 | 4E-3 | 3E-3 | 4E-3 | 3E-3 | 4E-3 | 3E-3 | 4E-3 | 3E-3 | 5E-3 | 4E-3 | 4E-3 | 4E-3 | 5E-3 | 4E-3 | 4E-3 | 4E-3 |

Table 5: Comparison of microphysical inversions based on simulated lidar data. The values shown come from evaluation of the root-mean-square and fractional metrics that are defined in eqs. 1 and 2 in the text. Cases are separated into fine, mixture and coarse mode cases. Different retrieval schemes are tested as are varying amounts of random uncertainty in the input optical data.



| Color coding scheme used in Tables 4 and 5 | | | |
| --- | --- | --- | --- |
| | Green | Yellow | Red |
| Effective Radius | 0-25% | 25-40% | >40% |
| Volume Concentration | 0-20% | 20-35% | >35% |
| Number Concentration | 0-75% | 75-125% | >125% |
| Real Part Refractive Index | 0.01-0.02 | 0.03-0.05 | >0.06 |
| Imaginary Part Refractive Index | <0.002 | 0.003-0.009 | >0.01 |

Table 6. Color coding scheme used for Tables 4 and 5. Green indicates values fully consistent with the desired uncertainties expressed in the ACE draft report, yellow indicates marginal consistency, while red indicates uncertainties inconsistent with desired uncertainties.



Figures

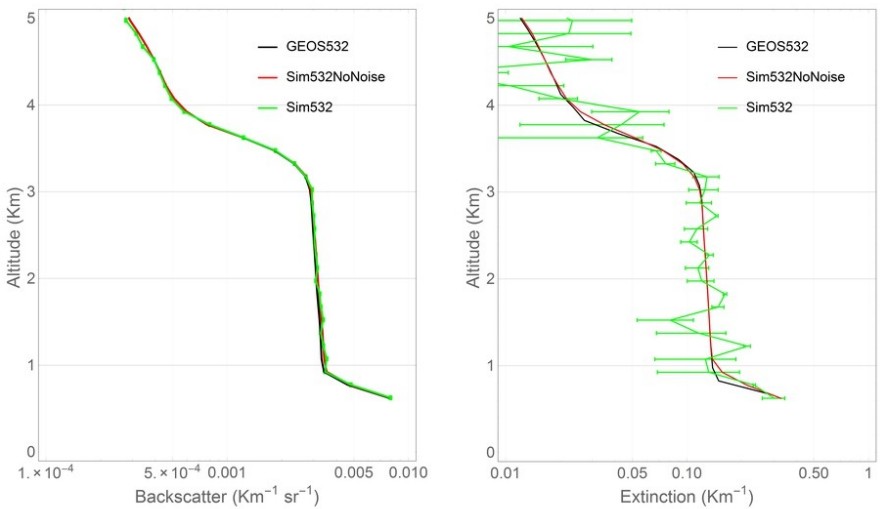

*Figure 1: Simulation of backscatter and extinction measurements from candidate multi-wavelength lidar system. Conditions simulated are from 0102 UT on July 24, 2009 track of the CALIPSO satellite while passing over central Africa. Orbital altitude used is 450 km. Horizontal resolution is 80 km. Vertical resolution is 150m for the backscatter and 450 m for the extinction.*





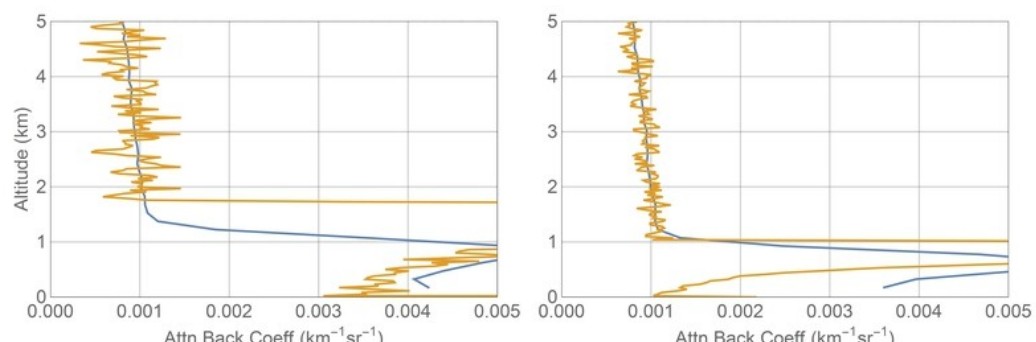

*Figure 2: Comparison of simulated ACE lidar (blue) and CALIPSO lidar (yellow) data during daytime (left @ 0.2 UT over the south Pacific ocean) and nighttime (right @ 1.2 UT off the west coast of Africa) on July 24, 2009.*




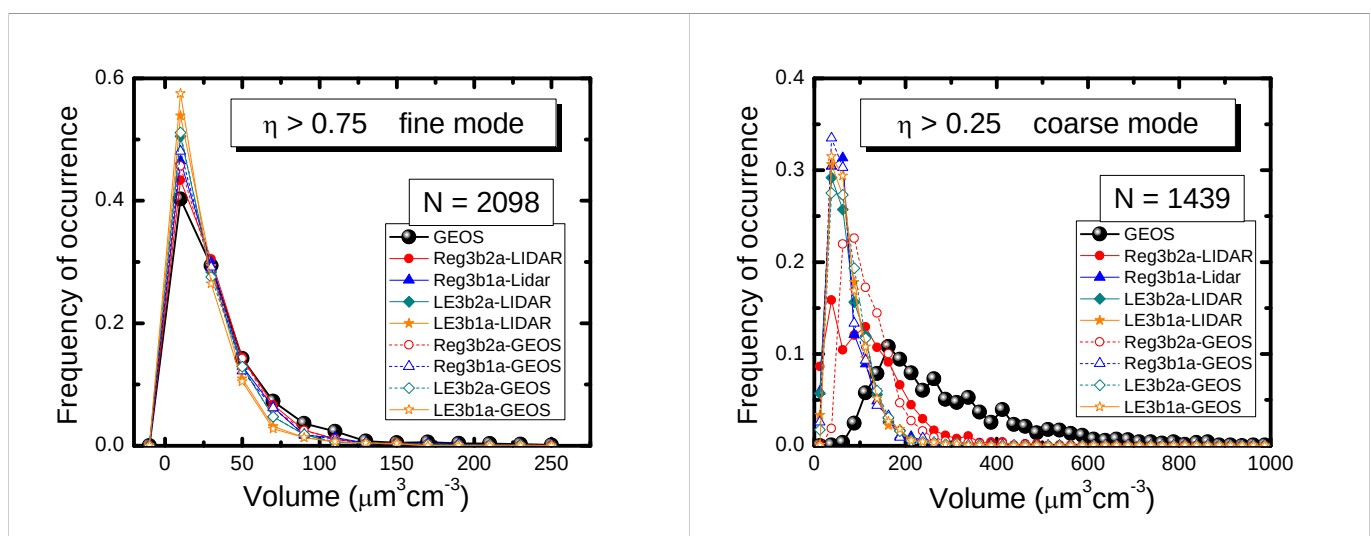

Figure 3. Comparison of aerosol volume from GEOS-5 and inversions using various optical data input and inversion schemes and selecting only for the 2098 qualifying fine mode aerosol cases (left) and the 1429 qualifying coarse mode cases (right) using the Spectral Deconvolution Algorithm. See text for description.



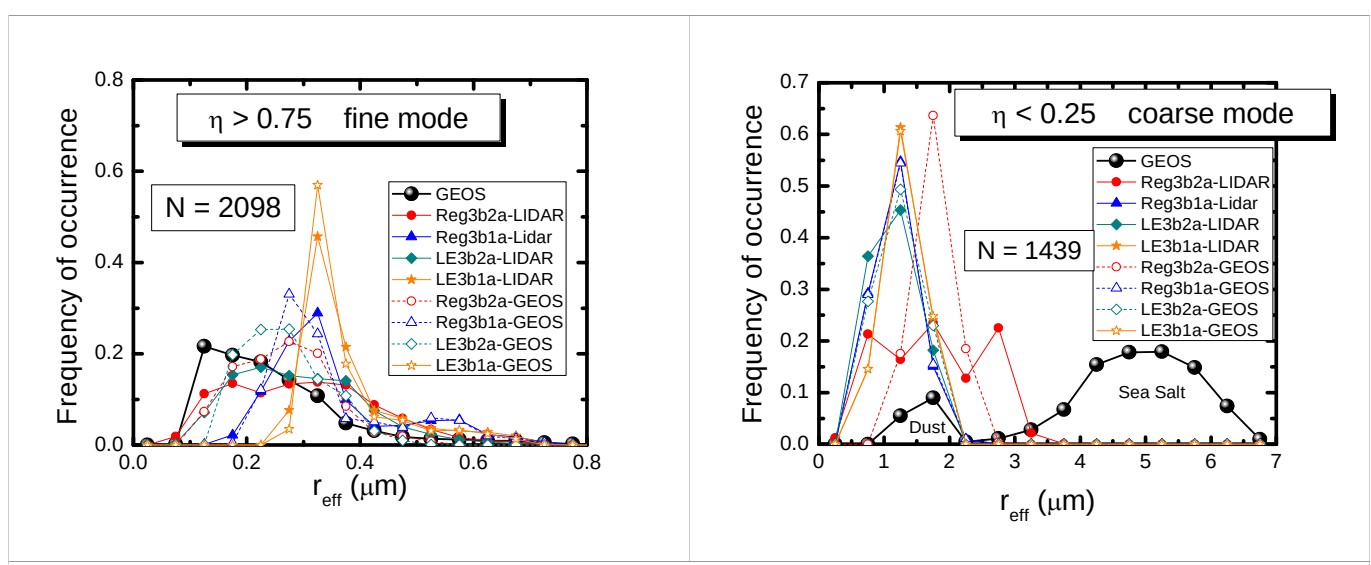

*Figure 4. Comparison of histograms of effective radius for the same fine (left) and coarse (right) mode cases used in Figure 3.*