# Peer review of "Simulations of spaceborne multiwavelength lidar measurements and retrievals of aerosol microphysics"

_Atmospheric Measurement Techniques, 2016_

## Referee Comment (RC1) · Anonymous Referee #1 · 25 Jul 2016

An interesting paper. I have a only mainly editorial concerns.

Line 13: What is meant by "reduced optical data"? It is likely not clear to the general reader and thus, a sentence explaining this should be inserted in the abstract. It is also unclear what is meant by "yield". I guess the authors are referring to the fraction of data volume that yields a final inversion product subject to some accuracy requirement.

Simulation approach: The part of the simulation chain going from the simulated HSRL measurements to the retrieved extinction and backscatter profiles seems, to me, to be missing. Perhaps, this aspect is well-covered by one or more of the references. However, even if this is the case, it needs to be explicitly called out and even a quick overview would improve the presentation.

[Figure]

Line 135: In the LE inversions were formulations using both volume and area based kernels used? Some studies have suggested that this can improve the accuracy of the volume and area retrievals respectively. Also were the full number of principle components, in general, used to generate the linear estimators?

Line 141: Why remove the uv and not the green? An, arbitrary decision or something else?

Line 158: Change "profile" to "profiles"

Line 188: What is the uv extinction the noisiest? Rayleigh backscatter and extinction is about 5 times higher in the uv and the green so I guess that the total (Rayleigh+aerosol) extinction can be more accurately determined than in the green. However, the contrast between the Raleigh and aerosol extinction is much greater at 532 nm and this leads to the final SNR being higher in the green than in the uv. Is this correct?

Figure 1: Why is the first data point at 600-700 meters? Is this a Topography effect or a retrieval issue?
* * *

---

## Referee Comment (RC2) · Anonymous Referee #2 · 10 Aug 2016

The authors present results of a simulation study on the retrieval of microphysical properties with two retrieval algorithms (inversion and linear estimation) that use data collected with a future space-borne multiwavelength HSRL. Such a lidar could be built in the context of the ACE (Aerosol, Clouds, Ecosystems) mission. For that purpose the authors use a global model (GEOS-5) that simulates optical properties of the atmosphere along a satellite orbit track that was chosen for this study. The simulations of lidar signals in addition allowed for the generation of optical data that can be used as input for the two algorithms that have been used for many years for the analysis of what is denoted as 3 backscatter (at 355, 532 and 1064 nm) and 2 extinction data (355 and 532 nm). The authors also consider the case of a reduced optical data set of 3

backscatter and 1 extinction (532 m) data sets. The authors provide information on the retrieval performance of the inversion of lidar-simulated data and optical data derived from simulations with GEOS-5. The authors introduce a threshold value of the minimum extinction coefficients (at 532 nm) in three atmospheric height layers. In that way they define a measure of quality of the optical data that are used for the inversion and the linear estimation algorithm. In addition the authors consider measurement errors of 15% of the optical data in their inversion and compare the retrieval quality to the results of inversion of error-free data. The authors derive effective radius and bulk parameters (number, volume, surface area) and refractive index.

It is encouraging to see that a team involved in lidar and algorithm development tackles the important question of assessing the potential of an "ACE" lidar with regard to providing vertically resolved microphysical aerosol properties. Thus the authors start a very important set of studies which will have impact on deciding if such a lidar can meet that the expectations regarding significantly improved, vertically resolved aerosol studies with space borne lidar in the 21st century.

I reject the paper in its current form. However, I strongly encourage the author to resubmit the paper. The idea of this study is highly important given the potential implications of any publication that impacts the decision making on what kind of HSRL should be built (3 backscatter and 2 extinction and one or more depolarization channels, or downscaled versions in which less wavelengths/channels might be used). I encourage the authors to carry out a more thorough analysis of their simulation study (with additional simulations where necessary) and particularly a thorough analysis of the implications of their study. The results presented here are not robust enough in order to allow for an informed assessment of the quality of the inversion results in view of the chosen lidar parameters.

The model (GEOS-5) that is used for the simulations is not well described in the paper. It remains a black box, comments are too general, and thus it is hard to judge how representative the aerosol optical data (generated for the study) are. Are the aerosol

scenarios (aerosol load, absorption properties, particle size) considered in this study representative? Particularly, the authors mention the aerosol model. I would like to see a critical description and assessment of this aerosol model. In how far is this aerosol model suitable for this study? What is the impact on the optical data generated in this study if this aerosol model (over)simplifies? The authors consider fine mode particles and coarse mode particles. Can coarse mode particles and their optical data modelled in a realistic way? Is dust considered in this study? If yes, are these optical data trustworthy in view of the fact that the modelling of optical data of dust (the authors mention Mie simulations) is questionable. There is plenty of literature on this topic. If this aerosol model has simplifications and constraints: these factors transfer to the input data set that are used for data inversion. Thus the inversion results may be biased or may not reflect the true situation in a reasonable way. What are the assumptions made in the simulations (with GEOS-5) of lidar profiles in general, aside from the aerosol model? Such information must be provided in a paper that wants to inform the reader on the potential benefit of a multi wavelength high spectral resolution lidar for aerosol studies in the 21st century. Further it is unclear what the quality of the microphysical data really is. The authors show a few tables which are compressing the information on the retrieval quality. It is impossible to decide how the optical data quality influences the retrieval results. Figures 3 and 4 show distribution functions, but these plots are too general and provide little insight on the real challenges one is faced with when experimental data are analyzed in a future ACE mission.

To my opinion the second main part of this paper is the generation of the optical data based on an atmospheric simulator. Thus it is equally important to provide sufficient information on the input data (simulation of lidar measurement and generation of optical data from GEOS-5). Why was one specific scenario chosen for this study? Is it representative? Could there be an unwanted bias with regard to the choice of this model. The choice of this example certainly influences the reader's opinion on the benefit of a lidar system with the parameters listed in Table 1. The authors mention that the lidar parameters were chosen on the basis of feedback from manufacturers and

lidar specialists (parameters are technically feasible and a reasonable representation): the tables with the results on the microphysical retrievals in part are not encouraging. What should be the technical parameters of such a lidar in order to obtain results on microphysics that meet the ACE requirements? Or is that more a problem of the lack of information content in the lidar data and not so much the quality of the optical data provided by such a lidar. Table 5 shows that the retrieval quality does not really change between input optical data with 0-15% error and 40-50% error (coarse mode, case C). I find little change in the retrieval quality for fine mode particles (case A and B) up to 40% random uncertainty. This result is of concern.

The simulated lidar profiles shown in Figure 1 provide little insight on the general situation. A more thorough overview on the lidar profiles including comments on the uncertainties (under different conditions along the flight track that was simulated) should be given. The authors show one(??) profile simulated for the HSRL lidar (specifications are given in table 1) and compare it to lidar profiles of the CALIPSO lidar (in terms of how much better the HSRL lidar could be). I think such a comparison should be made more thorough as it also touches the topic of the benefit of this new lidar compared to the CALIPSO lidar. In addition we need to keep in mind that data from CALIPSO cannot be used for the inversion algorithm that is used by the authors. The authors mention vertical resolution of 150 m for the backscatter and 450 m for the extinction profiles. Did they use the same vertical resolution (backscatter and extinction profiles) for generating the optical input data that are used in the inversion?

Tables 2 - 4 are helpful as they allow for a first insight on the potential yield of data sets that can (from the total set of data points collected during a track) be used for data inversion; please check Table 6 (color coding information for surface concentration is missing). This yield, i.e. the number of data points suitable for inversion are based on the constraint of extinction coefficient at 532 nm (0.02 km1-1). Do other factors play a role in that yield? How big (number of data points suitable for inversion) would the yield be if extinction is set to 0.01 km-1? What about the quality of backscatter coefficients?

That does not play a role? The authors decide to remove extinction at 355 nm and run another set of simulations. This is a good idea as it allows for some insight if a fully blown 3 backscatter and 2 extinction lidar is not used. Why not do the same study by removing extinction at 532 nm? What would be the yield of data points if you prescribe a constraint on the extinction coefficient at 355 nm? What would be a suitable threshold value?

---

## Author Comment (AC1) · 6 Sep 2016

Please see the attached supplement for complete responses to both reviewers comments along with revised manuscript.

Please also note the supplement to this comment:
http://www.atmos-meas-tech-discuss.net/amt-2016-174/amt-2016-174-AC1-supplement.zip